# The Characteristics of Transparent Non-Volatile Memory Devices Employing Si-Rich SiO_X_ as a Charge Trapping Layer and Indium-Tin-Zinc-Oxide

**DOI:** 10.3390/nano9050784

**Published:** 2019-05-22

**Authors:** Joong-Hyun Park, Myung-Hun Shin, Jun-Sin Yi

**Affiliations:** 1School of Electronic Electrical Engineering, Sungkyunkwan University, Suwon 440-746, Korea; jhyun21.park@gmail.com; 2School of Electronics and Information Engineering, Korea Aerospace University, Goyang-City 412-791, Korea

**Keywords:** ITZO, transparent, NVM, charge trapping, memory window, retention

## Abstract

We fabricated the transparent non-volatile memory (NVM) of a bottom gate thin film transistor (TFT) for the integrated logic devices of display applications. The NVM TFT utilized indium–tin–zinc–oxide (ITZO) as an active channel layer and multi-oxide structure of SiO_2_ (blocking layer)/Si-rich SiO_X_ (charge trapping layer)/SiO_X_N_Y_ (tunneling layer) as a gate insulator. The insulators were deposited using inductive coupled plasma chemical vapor deposition, and during the deposition, the trap states of the Si-rich SiOx charge trapping layer could be controlled to widen the memory window with the gas ratio (GR) of SiH_4_:N_2_O, which was confirmed by fourier transform infrared spectroscopy (FT-IR). We fabricated the metal–insulator–silicon (MIS) capacitors of the insulator structures on n-type Si substrate and demonstrated that the hysteresis capacitive curves of the MIS capacitors were a function of sweep voltage and trap density (or GR). At the GR6 (SiH_4_:N_2_O = 30:5), the MIS capacitor exhibited the widest memory window; the flat band voltage (Δ*V*_FB_) shifts of 4.45 V was obtained at the sweep voltage of ±11 V for 10 s, and it was expected to maintain ~71% of the initial value after 10 years. Using the Si-rich SiO_X_ charge trapping layer deposited at the GR6 condition, we fabricated a bottom gate ITZO NVM TFT showing excellent drain current to gate voltage transfer characteristics. The field-effect mobility of 27.2 cm^2^/Vs, threshold voltage of 0.15 V, subthreshold swing of 0.17 V/dec, and on/off current ratio of 7.57 × 10^7^ were obtained at the initial sweep of the devices. As an NVM, Δ*V*_FB_ was shifted by 2.08 V in the programing mode with a positive gate voltage pulse of 11 V and 1 μs. The Δ*V*_FB_ was returned to the pristine condition with a negative voltage pulse of −1 V and 1 μs under a 400–700 nm light illumination of ~10 mWcm^−2^ in erasing mode, when the light excites the electrons to escape from the charge trapping layer. Using this operation condition, ~90% (1.87 V) of initial Δ*V*_FB_ (2.08 V) was expected to be retained over 10 years. The developed transparent NVM using Si-rich SiOx and ITZO can be a promising candidate for future display devices integrating logic devices on panels.

## 1. Introduction

The ionic bonded amorphous oxide semiconductor (AOS) materials with a large optical bandgap were introduced for the applications of transparent electrode like indium tin oxide (ITO) or zinc oxide (ZnO) [1]. Nowadays, AOS have been used in active matrix devices of thin film transistor (TFT). The conduction band of the AOS is related with the ns orbitals of heavy metals, such as indium (In), gallium (Ga) and zinc (Zn). Even when AOS is composed in an amorphous phase, electrons can easily transport through the ns orbitals because the ns orbitals of AOS are overlapped by large size metals and are less sensitive to the arrangement of atoms [2]. The AOS TFT has many advantages compared with an amorphous silicon (Si) based TFT. It has high field-effect mobility (~10 cm^2^/Vs) and very low leakage current (~10^−12^ A/cm^2^) and it can be very uniformly fabricated even on flexible substrates [3]. Moreover, AOS can be applicable for transparent and flexible display devices because it has a high optical bandgap (>3 eV) and it can be made at low process temperature (~200 °C) [4]. In spite of a short period of research and developments, it can be used in the manufacturing of commercial display products for active matrix organic light emitting diodes (AMOLED) [5].

In a future display, various active components such as pixel devices, gate and source driver, logic devices and non-volatile memory (NVM) will be integrated on a display panel. The basic operating of NVM devices is to store charges in a dielectric layer; when charges are trapped or detrapped in the dielectric layer of an NVM transistor, the threshold voltage (*V*_TH_) of the transistor can be altered between two states as programming and erasing states. Two representative types of NVM in recent studies are floating gate and charge trapping layer memories [6,7]. In the floating gate NVM, the charges injected from a channel region through a tunneling layer are stored in a floating gate. The charge trapping NVM consists of three important layers: tunneling, charge trapping and blocking layers, where the charges are injected from a channel through the thin tunneling layer (a dielectric layer near the channel) and stored in the charge trapping layer. The thickness and interface traps of the tunnel layer are critical because they directly affect the electric field in the tunnel layer, which determines the tunnel current. The charge trapping layer stores the charges in the traps of the layer; as many shallow or deep traps, as possible are favorable. The blocking layer near the gate should be thick enough to block charge transport to the gate region, and the thickness and the trap states of the blocking layer must be carefully designed because they can also affect the electric field in the tunnel layer.

The AOS based charge trapping NVMs have been researched in previous literature. Zhang et al. fabricated a novel multi-level NVM using an amorphous Indium-Gallium-Zinc-Oxide (a-IGZO) layer as both storage and channel layers at low temperatures [8] but reported the programming/erasing (P/E) characteristics with such drawbacks as poor data retention, insufficient mobility (~2 cm^2^/Vs), and high erasing voltage (−15 V). Bak et al. reported on a-IGZO NVMs using an IGZO channel and ZnO trap layers fabricated at a low process temperature of 200 °C [9], which also exhibited high programming/erase voltages (20 V/−20 V), lower mobility (0.2 cm^2^/Vs), hump in transfer curve, and poor data retention. Yin et al. reported on an NVM with a bottom gate n-type IGZO TFT structure [10,11] of IGZO (channel)/Al_2_O_3_ (tunneling layer)/IGZO (charge trapping layer)/Al_2_O_3_ (blocking layer)/Mo (bottom gate) layers; low *V*_TH_ of 0.7 V and high field-effect mobility of 10.3 cm^2^/Vs are exhibited, but reliability characteristics were still so poor that the program/erase cycles and retention time were less than 10^3^ times and 10^3^ s, respectively. The IGZO and its interface were not optimized yet as a charge trapping layer so that the stored charges easily escaped. 

In this work, we fabricated a charge trapping NVM employing gallium free indium tin zinc oxide (ITZO) as an active channel layer of NVM and Si-rich oxide (SiO_X_) as a charge trapping layer; the bottom gate ITZO NVM with SiO_2_ (blocking layer)/SiO_X_ (charge trapping layer)/SiO_X_N_Y_ (tunneling layer) was fabricated. The ITZO layer has many advantages in that it reported a high field-effect mobility (>30 cm^2^/Vs) and high transparency [12,13]. The high field-effect mobility of the ITZO is useful for high density panels, such as virtual reality displays. Though the oxygen vacancies in the bulk and interface of the AOS channel might shift the *V*_TH_ under a bias voltage, these traps can be easily released by source-drain voltage, so that they will not influence on the long-term retention characteristics of NVM devices. The Si-rich SiO_X_ layer has many Si phases and defect sources in the SiO_X_ matrix, such as Si-H bonds in HSi_3_O, HSi_2_O_2_, H_2_SiO_2_, so that it can enlarge memory windows, which means it has charge storage ability at a certain applied bias voltage. Moreover, the Si-rich SiO_X_ layer has low operating voltage, which can save the power consumption of a memory device. In addition, the charge states of the Si-rich SiO_X_ layer can be experimentally controlled or can be engineered and optimized by simply controlling a gas ratio of silane and nitrous oxide (SiH_4_:N_2_O) during the deposition process. Thus, the Si-rich SiO_X_ can be a very promising material as a charge trapping layer. The increase of states in the charge trapping layer was investigated by a fourier transform infrared (FT-IR) spectroscopy, and the expansion of the memory window was confirmed by capacitive-voltage (*C-V*) measurement. The superior reliability characteristics of the fabricated bottom gate ITZO NVM was presented. This work will be useful to develop the next transparent future display integrating of display components.

## 2. Device Fabrication

The schematics of fabricated devices are shown in Figure 1. To characterize and optimize the Si-rich SiO_x_ layer, metal–insulator–silicon (MIS) capacitors were fabricated as shown in Figure 1a; the stack of aluminum/silicon dioxide (SiO_2_)/Si-rich SiO_X/_silicon oxynitride (SiO_X_N_Y_) layer was deposited by inductive coupled plasma chemical vapor deposition (ICP-CVD) on a n-type single crystalline Si substrate of about 1–5 Ωcm resistivity. At first, a 2.7 nm-thick SiO_X_N_Y_ tunneling layer was formed using a N_2_O plasma treatment under a N_2_O gas flow of 2.7 sccm for 8 min at 170 °C. A 20 nm-thick SiO_X_ charge trapping layer was deposited with different SiH_4_:N_2_O gas ratio changed from 1:2 to 6:1, and a 20 nm-thick SiO_2_ blocking layer was deposited under the SiH_4_:N_2_O gas ratio of 2:60 at the same temperature. The insulator stack was annealed using a rapid thermal annealing at 250 °C for 30 min. The layer thicknesses of the device are very important for the proper operation. Especially, the thickness of the blocking layer can directly affect the internal electric field and the amount of the trapped charges. We selected the thickness of each layer based on those that were optimized through previous experiments [14]. The detailed deposition condition for MIS capacitor was summarized at Table 1. Finally, a silver (Ag) electrode of 100 nm was deposited at 350 °C by a thermal evaporation method. The thickness of each layer was measured by an ellipsometer. The fabricated MIS capacitor was measured by an impedance analyzer at high frequency of 1 MHz.

Using the optimized Si-rich SiO_x_ charge trapping layer, a bottom gate charge trapping NVM with ITZO layer was fabricated as shown in Figure 1b. The stack of insulator layers was deposited by ICP-CVD on the n-type single crystalline Si substrate of a gate electrode. The SiO_2_, SiO_X_, SiO_X_N_Y_ layers were deposited sequentially in ICP-CVD. The deposition conditions of the insulator layers were the same as those of the MIS capacitor. On top of the insulator stack, a 25 nm-thick ITZO active layer was deposited by DC magnetron sputtering method at room temperature; the ITZO target was composed of In_2_O_3_:SnO_2_:ZnO (=1:1:1), the RF power for the plasma deposition was 80 W, and the working pressure was 5 mTorr with an oxygen partial pressure of 25–30%. After annealing at 250 °C to improve the device performance, a 150-nm thick Ag layer for source/drain electrodes was deposited and patterned by photo lithography. The current-voltage and capacitive-voltage characteristics for the fabricated devices were measured by a semiconductor parameter analyzer (EL420C) and an impedance analyzer (Hewlett Packard 4192A LF), respectively.

## 3. Results & Discussions

Figure 2 shows the measured results of FT-IR transmittance spectroscopy (Bruker, IFS-66/S) for the SiO_X_ charge trapping layer with a different SiH_4_:N_2_O gas ratio to evaluate as a charge trapping layer. The SiO_X_ layers showed the high absorption of Si-O bending peaks (or defects known as non-bridging oxygen hole center, NBOHC) at 860 cm^−1^ [15]. These NBOHC defects are known as the trapping sites to retain the injected charges. The charge trapping sites are observed in the all the conditions. In particular, SiO_x_ having a high Si content shows a remarkably large number of Si-H bonds as well as HSi_3_O, HSi_2_O_2_, H_2_SiO_2_, and HSiO_3_ at around 2000~2300 cm^−1^ [16,17], and as the GR increases, the absorptions around 2000–2300 cm^−1^ also increase. This implies that many Si phases and defects can exist in the Si-rich SiO_X_. The numerous Si phases are desirable in a charge trapping layer because of their potential for large capacity and wide memory window, so the Si-rich SiO_X_ grown at the gas ratio of 6:1 is preferable. 

The charging effects in conventional MIS structures are usually attributed to the traps in bulk of insulator and traps in the interface between the insulator and semiconductor. Each charge trapping effect can be distinguished by capacitance-voltage (*C*-*V*) measurement. To compare the charge storage characteristics of the NVMs for the charge storage layers grown by different SiH_4_:N_2_O GR conditions, the *C*-*V* characteristics of the MIS capacitors were measured at forward and reverse sweeping modes because each MIS capacitor has the same insulator structure of the NVM. In the case of the SiO_X_ film using GR0.5, the charge trapping effect due to bulk trap in the insulator was high and the charge trapping effect due to the interface trap between insulator and semiconductor was almost zero.

Figure 3 shows measured *C*-*V* characteristics of the MIS capacitor with the insulator stack of SiO_2_/SiO_X_/SiO_X_N_Y_. A small voltage signal of 1 MHz was applied between top (Ag) electrode and bottom n-type single crystalline Si substrate with the bias voltage sweeping from depletion (−V) to accumulation (+V) for the capacitance measurement. At the positive bias voltage, electrons, the majority carriers of Si channel were accumulated at the interface to the SiO_X_N_Y_, the MIS capacitance was close to that of the overall insulator capacitor of SiO_2_/SiO_X_/SiO_X_N_Y_ (C_OX_, ~8.48 × 10^−8^ F/cm^2^). At the negative bias voltage, the electrons near the SiO_X_N_Y_ were repelled and the depletion region was expanded until the inverted (p-type) channel was formed in the n-type Si region, so that the device capacitance becomes the effective value of series-connected capacitors made of the insulator capacitor, SiO_2_/SiO_X_/SiO_X_N_Y_ and the depletion capacitor in the n-type Si substrate. When the bias voltage of the MIS capacitor was swept, the *C*-*V* characteristic curves were changed for the sweeping voltage range and the sweeping mode: forward (from negative to positive) and reverse (from positive to negative) mode. In the forward mode, part of the accumulated electrons at the interface to the SiO_X_N_Y_ was trapped in the trapping layer via tunneling process, which kept the interface region of the n-type Si depleted even after the applied voltage was removed. In the reverse mode, the electrons in the trapping layer escaped to the n-type Si; the empty trap states (or holes occupied) induce the electrons in the n-type Si to reduce the depletion region. Therefore, the hysteresis curves are clockwise in Figure 3. 

The number of tunneled carriers depends both on the electric field intensity in the tunneling layer, and the amount of available states in the trapping layer. As the sweep voltage range increases from −5–5 V to −15–15 V, the flat band voltage (*V*_FB_) shifts due to electron or hole trappings also increasing. The flat band voltage at which population inversion occurred in the n-type Si (or channel) was determined by the metal-silicon work function difference and the various oxide charges. The total hysteresis can be presented the sum of the difference of the *V*_FB_ shift (Δ*V*_FB_) by the hole and electron charging. Though the *V*_FB_ of real MIS structures was further affected by the presence of charge in the oxide or at the oxide-semiconductor interface, for convenience, *V*_FB_ was measured at middle of *C-V* curve in this paper.

When the charge trapping layer was deposited at the condition of GR0.5, the Δ*V*_FB_ was imperceptible even at the sweeping voltage of −15~15 V, and, the Δ*V*_FB_ from characteristic *C*-*V* curves was very small. It means that GR0.5 is not suitable for a charge trapping layer. When the charge trapping layer was deposited by GR2, Δ*V*_FB_ was 7.31 V in the large range voltage sweeps from −10 V to +10 V, but the Δ*V*_FB_ at the small sweep voltage range of −5–5 V is still imperceptible. This was attributed to small voltages not being able to give rise to a considerable charging effect in the defect sites of Si-rich SiO_X_ film. At the condition of GR6, wide Δ*V*_FB_ were exhibited as 10.92 and 2.47 V at the sweep voltage range of −10–10 and −5–5 V, respectively. As shown in Figure 2, the charge trapping layer with more Si phases can increase the Δ*V*_FB_ (memory window) to more clearly distinguish memory states. The SiO_X_ layer grown at the highest SiH_4_:N_2_O gas ratio (GR6) condition can capture the greatest amount of charge with a large number of Si phases.

We compared the charge storage performance of the fabricated MIS capacitor for the GR condition of the charge trapping layer using the measured *C*-*V* characteristics. Figure 4a,b shows the operation voltage of programming/erasing as a function of the bias voltage comparing GR2 and GR6. In both conditions, the Δ*V*_FB_ also increased with the applied bias voltage; the internal electrical field in the MIS capacitor increased with the bias voltage, which could facilitate the tunneling of the electrons, and make the electrons easily trapped or detrapped in the charge trapping layer. But, the difference of Δ*V*_FB_ was observed. At the bias voltage of ±11 V, the Δ*V*_FB_ were 2.18 and 4.45 V for the charge trapping layers of GR2 and GR6. The difference of Δ*V*_FB_ was due to the difference of charge trapping density as shown in FT-IR spectroscopy in Figure 2; a higher trap density in the Si-rich SiO_X_ film can be obtained by the deposition under higher GR.

Figure 4c,d shows charge retention characteristics for the MIS capacitors of the different GR, which is induced from the *C*-*V* measurements. After applying the bias voltages of programming (11 V) and erasing (−11 V) to the MIS capacitors for 10 s, *C*-*V* characteristics were measured over time to extract Δ*V*_FB_ between the programming and erasing states. The initial Δ*V*_FB_ were 3.07 and 4.24 V for GR2 and GR6 MIS capacitors. The Δ*V*_FB_ was gradually degraded during the observation for 10^3^ s. Assuming the same Δ*V*_FB_ deterioration trend, Δ*V*_FB_ was estimated to hold ~67% and ~71% of the initial Δ*V*_FB_ of GR2 and GR6 MIS capacitor for 10^9^ s (nearly 10 years). Assuming that the NVM device was working 10 years, the retention of GR2 and GR6 was about 67% and 71% of each initial Δ*V*_FB_, respectively. Additionally, a low operation voltage of about ±11 V was used. These good retention properties are promising for NVM because of using the SiO_2_/SiO_X_/SiO_X_N_Y_ memory stack.

Figure 5 shows the drain current to gate voltage (*I*_D_-*V*_GS_) transfer characteristics of the fabricated metal/SiO_2_/SiO_X_/SiO_X_N_Y_/ITZO NVMs at programming and erasing modes, which is measured at the drain voltage of 1 V. The fabricated device exhibited excellent transistor characteristics because of the high field-effect mobility (*μ*_FE_) of ITZO layer. The measured initial values of *μ*_FE_, *V*_TH_, subthreshold swing (S.S.), and on/off current ratio were 27.2 cm^2^/Vs, 0.15 V, 0.17 V/dec and 7.57 × 10^7^, respectively. At the program mode of the NVMs, the positive-voltage pulses of 1 μs in the voltage range of 9–13 V were applied to the gate, and the *I*_D_-*V*_G_ transfer curves were shifted to the positive. When the positive voltage pulse was applied to the gate, an internal electric field was applied to the SiO_2_/SiO_X_/SiO_X_N_Y_ dielectric stack. As a result, electrons could be injected from the conduction band of the ITZO channel through the tunneling layer and stored in the Si-rich SiO_X_ charge trapping sites. It is known that the electrons were trapped not only in shallow or deep traps of SiO_X_, but also in OH¯, H_+_ residues, and oxygen vacancy (V_O_) sites at the interfaces with the ITZO channel layer in the programming mode [18]. Thus, the memory window can be expected to increase due to the plenty of the negative charges. The Δ*V*_TH_ by the gate voltage pulses of 11 and 13 V were 2 and 3.7 V, respectively. Higher programing voltage and more states both in shallow or deep traps of the charge trapping layers add more advantages to the NVMs because the tunneling electric field can be higher with the gate voltage, and more electrons can be stored in the trap states as in the MIS capacitors. 

Various tunneling mechanisms have been suggested, such as direct tunneling, Fowler–Nordheim (FN) tunneling and channel hot electron (CHE) tunneling processes [19]. In our work, an ultra-thin tunneling layer of less than 4 nm was used, and the energy barrier (*Φ*_B_) between ITZO and SiO_X_N_Y_ was ~2 eV, so we expected that a direct tunneling process could be dominantly involved. We noted that specifying one of these processes was difficult because these processes can occur together. So, it can be done in future studies.

A transparent NVM can be programmed with the positive-voltage pulses at the gate with plentiful negative charges sources but erasing at the negative-voltage pulses at the gate cannot easily erase the NVMs. Even with the significant negative gate voltage (−20 V), the Δ*V*_TH_ did not move back to the negative direction, which was a very different result from the MIS capacitor. The erasing behavior in the AOS NVM was introduced [20]. In the MIS on n-type Si substrate, the bandgap of Si was smaller than that of the Si-rich SiOx (~2.4 eV), so the trapped electrons could be easily tunneled to the conduction band of the n-type Si when the negative bias voltage bends the conduction band down. However, the bandgap of ITZO (~3.2 eV) was higher than that of Si-rich SiOx, and electrons could not tunnel to the inner state of the forbidden band (in the bandgap) of the ITZO channel at a low negative bias voltage. A possible reason for this result is that the AOS, such as IGZO and ITZO, has only an electron carrier in nature. When the negative voltage was applied in Si for the erasing operation, hole carriers were injected to the charge trapping layer through the tunneling layer. Then, injected electrons could be neutralized by the hole carrier. However, few hole carriers were available in erasing behavior. Thus, the *V*_TH_ in ITZO NVM was very difficult to shift back to the negative side. 

As shown in Figure 6, the injected electrons were detrapped from the programmed electrons using white light having a wavelength from 400 to 700 nm with 10 mWcm^−2^ and negative gate bias of −1 V for 1 s. As a result, the transfer curve returned to the pristine value. It was also reported that there was charge detrapping behavior under the light-assisted erasing process [21]. To investigate light induced erasing behavior, the optical bandgap was measured by UV-visible spectra. The optical bandgap of ITZO and SiO_X_N_Y_, SiO_X_, was 3.05~3.15 eV, 5.1 eV and 2.3 eV, respectively. To explain erasing behavior, Figure 6 shows a band diagram of the Metal/SiO_2_/SiO_X_/SiO_X_N_Y_/ITZO NVM under white light irradiation. When the white light was irradiated on the ITZO layer, the electron hole pair was generated. The negative gate voltage made the electric field so that holes were injected to the charge storage layer through the tunneling layer at the valance band of ITZO. 

Because the low bandgap of the Si-rich SiO_X_ and very thin tunneling layer was used, it was expected that its low valence band offset was aligned with ITZO layer [22,23]. Thus, generated holes were easily trapped under light illumination at the SiO_X_/SiO_X_N_Y_ interface or at SiO_X_ bulk trap states in the storage layer. Injected holes were recombined with trapped electrons, and neutralized. Thus, *V*_TH_ in the fabricated NVM could return to the pristine value. Surplus electrons by white light in ITZO became carriers, so that carrier concentration as well as conductivity of ITZO layer was increased (*σ* = qμn). It can move *V*_TH_ shift to negative direction like as a negative bias illumination stress (NBIS) condition [24].

We need to note that the control of external light illumination must be considered to use the ITZO NVM devices properly. For the application of transparent NVM, in-depth research of erasing electrons will be required, such as integration with light emission devices, which can trigger the erasing operation. A bandgap engineering for the layer structure, optimizing the conduction/valance band offset between the tunneling layer and the charge trapping layer, can reduce tunnel barrier and improve the erasing operation, or the excitations of the trapped charges which can be done by local thermal heating or electro-magnetic resonating terminals can promote the tunneling process without the external light source.

The retention characteristics of the fabricated Metal/SiO_2_/SiO_X_/SiO_X_N_Y_/ITZO NVM at programming state are shown in Figure 7. After applying programming voltage pulse of 11 V and 1 μs, the threshold voltage shift was measured for 10^4^ s. From the results of Figure 7, it is expected that the shifted threshold voltage can be maintained for 10 years. The initial memory window was ~2.08 V, and after 10^4^ s, the memory window was maintained as ~1.95 V and was expected to be ~1.87 V after 10 years; ~90% of the trapped charges can be retained for ~10 years.

## 4. Conclusions

A charge trapping NVM was fabricated for the logic devices integrated on display panels. The bottom gate NVM consisted of ITZO as an active channel and multi-insulator gate structure of SiO_2_ (blocking layer)/Si-rich SiO_X_ (charge trapping layer)/SiO_X_N_Y_ (tunneling layer). The insulator gate structure was deposited using a ICP-CVD, and by changing the GR of SiH_4_:N_2_O, the trap states of Si-rich SiOx charge trapping layer were controlled. The MIS capacitors using the insulator structure on n-type Si substrate shows clear hysteresis capacitive curves as a function of sweep voltage and trap density (or GR). When the SiOx charge trapping layer of GR6 was used, the Δ*V*_FB_ of the MIS capacitor with the SiOx charge trapping layer was 4.45 V at the sweep voltage of ±11 V and was expected to maintain ~71% after 10 years. The NVM TFT with the same insulator gate structure and ITZO channel exhibited good *I*_D_-*V*_G_ transfer characteristics because of high field-effect mobility (*μ*_FE_) of ITZO; the initial μ_FE_, *V*_TH_, subthreshold swing (*S.S.*), and on/off current ratio were 27.2 cm^2^/Vs, 0.15 V, 0.17 V/dec and 7.57 × 10^7^, respectively. In a programing mode by positive gate voltage pulses of 11 V and 1 μs, the Δ*V*_TH_ of the NVM TFT was shifted by 2.08 V. In erasing mode, the Δ*V*_TH_ returned by negative voltage pulses of −1 V and 1 μs under the illumination of 400–700 nm light with ~10 mWcm^−^^2^ intensity power. The illuminated light excited the electrons in the charge trapping layer, and assisted the electrons to escape and tunnel in the erasing mode. Using this operation condition, ~90% (1.87 V) of initial Δ*V*_TH_ (2.08 V) of the NVM TFT was expected to be retained over 10 years. These results are of superior reliability and characteristics for a bottom gate ITZO NVM. The developed device and work can be used in future transparent display embedding and in the integrating of circuit components on display panels, and further research on easing the electrons without light assistance will be useful for the practical usage of the ITZO NVM. 

## Figures and Tables

**Figure 1 nanomaterials-09-00784-f001:**
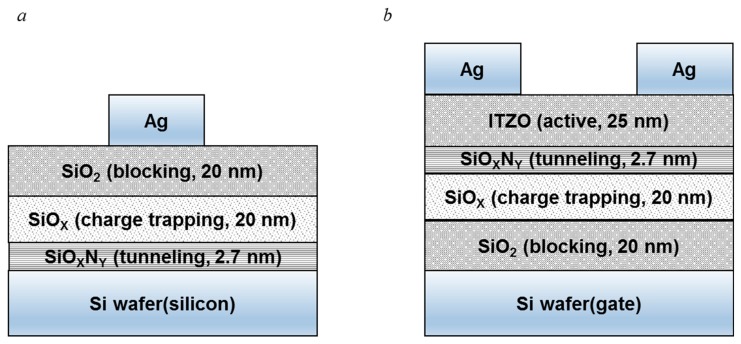
Schematic cross sections of (**a**) metal–insulator–silicon (MIS) (Ag/SiO_2_/SiO_X_/SiO_X_N_Y_/Si) capacitor, and (**b**) indium tin zinc oxide (ITZO) non-volatile memory (NVM) with a SiO_2_/SiO_X_/SiO_X_N_Y_ stack.

**Figure 2 nanomaterials-09-00784-f002:**
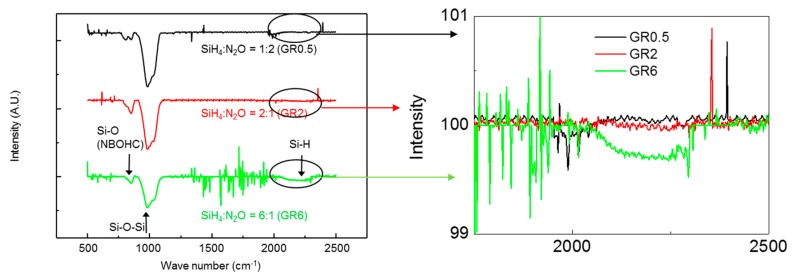
The FT-IR transmittance spectroscopy of the SiO_X_ layer as a charge trapping layer with different SiH_4_:N_2_O gas ratio (GR).

**Figure 3 nanomaterials-09-00784-f003:**
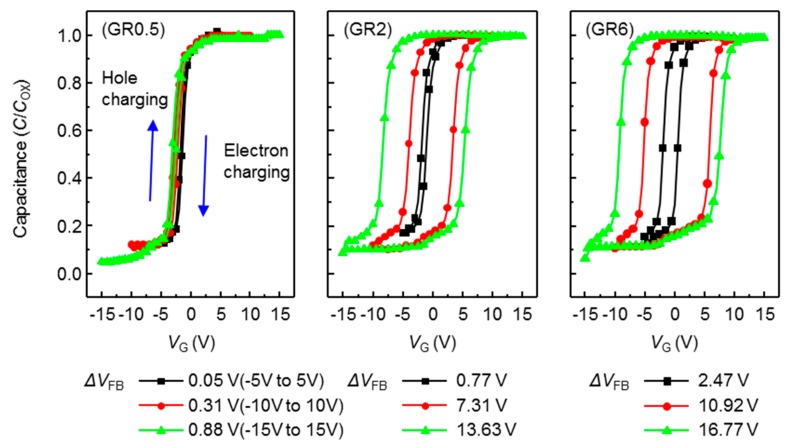
The capacitance-voltage characteristics curves of the Metal/SiO_2_/SiO_X_/SiO_X_N_Y_/Silicon capacitor with different SiH_4_:N_2_O gas ratio for Si-rich SiOx charge trapping layer; the flat band voltage shift (Δ*V*_FB_) (sweep voltage range) are 0.05 V (−5–5 V), 0.31 V (−10–10 V), and 0.88 V (−15–15 V) for black, red, and green lines, respectively.

**Figure 4 nanomaterials-09-00784-f004:**
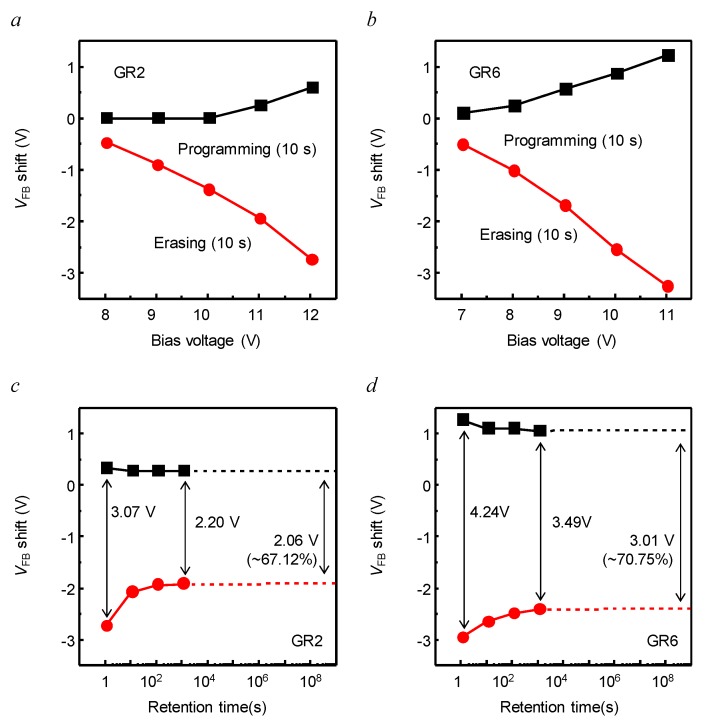
The operation voltages of programming/erasing charges trapped in MIS capacitors for bias voltage with different SiH_4_:N_2_O gas ratio for Si-rich SiO_X_ charge trapping layer: (**a**) GR2 and (**b**) GR6, and extrapolated retention characteristics up to 10 years (10^9^ s) of the Metal/SiO_2_/SiO_X_/SiO_X_N_Y_/Si capacitor with different SiH_4_:N_2_O gas ratio for Si-rich SiO_X_ charge trapping layer: (**c**) GR2 and (**d**) GR6.

**Figure 5 nanomaterials-09-00784-f005:**
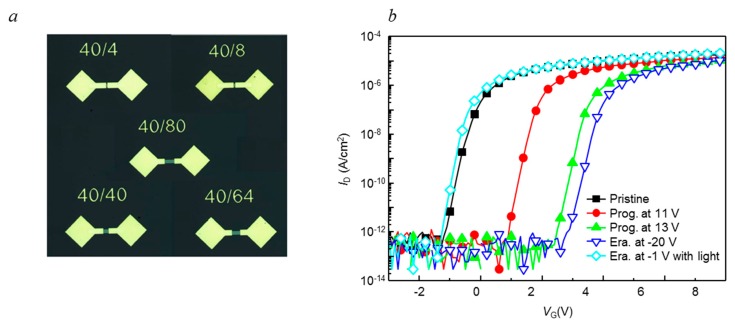
(**a**) Optical microscopy image of the fabricated ITZO NVM and (**b**) Programming and erasing characteristics of the Metal/SiO_2_/SiO_X_/SiO_X_N_Y_/ITZO NVM.

**Figure 6 nanomaterials-09-00784-f006:**
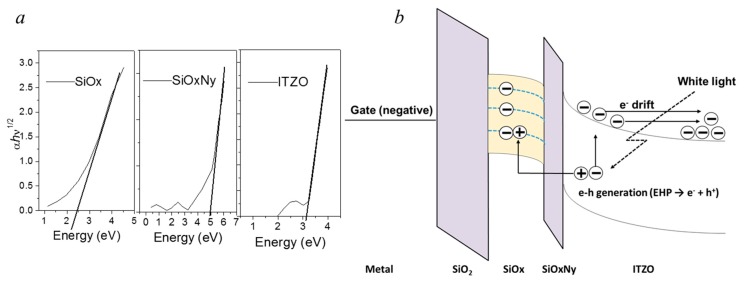
(**a**) The UV-vis spectroscopy and (**b**) Band diagram of the Metal/SiO_2_/SiO_X_/SiO_X_N_Y_/ITZO NVM under the white light irradiation.

**Figure 7 nanomaterials-09-00784-f007:**
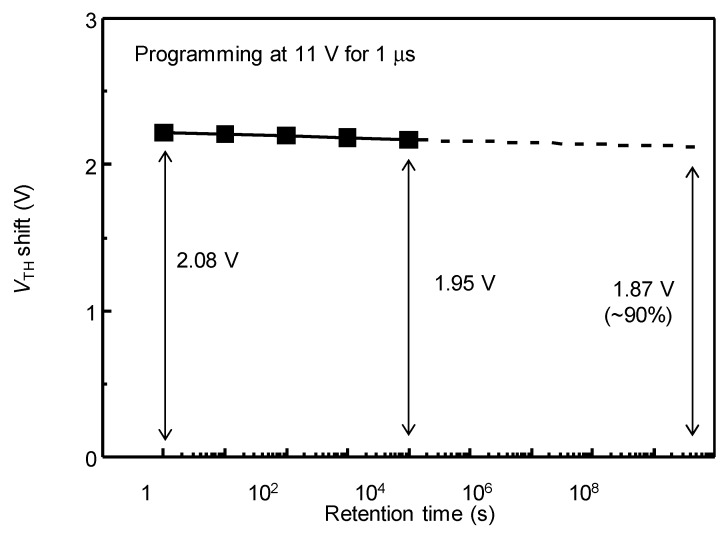
The extrapolated retention characteristics up to 10 years (10^9^ s) of the Metal/SiO_2_/SiO_X_/SiO_X_N_Y_/ITZO NVM.

**Table 1 nanomaterials-09-00784-t001:** Deposition condition for deposition of the SiO_2_/SiO_X_/SiO_X_N_Y_ layer.

	SiH_4_ (sccm)	N_2_O (sccm)	GR (*)	Temp (°C)	Pressure (mTorr)	Power (W)	Thickness (nm)
**SiO_2_ (blocking)**	2	60		170	100	25	20
**SiO_X_ (charge tapping)**	5	10	0.5				
	10	5	2	170	100	25	20
	30	5	6				
**SiO_X_N_Y_ (tunneling)**		2.7		170	10	25	2.7

* GR: gas ratio of SiH_4_:N_2_O.

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
