# Peer review of "The Characteristics of Transparent Non-Volatile Memory Devices Employing Si-Rich SiOX as a Charge Trapping Layer and Indium-Tin-Zinc-Oxide"

_nanomaterials, 2019, doi:10.3390/nano9050784_

Round 1
Reviewer 1 Report
See attached review.

Author Response
The writing is reasonable; however, there are typos, grammatical errors, missing words, and poor word choices throughout. It makes the paper harder to read. The paper should be proofread thoroughly and increased in clarity.
Thanks for kind suggestion. We check them again before submission of the revision. We think that this manuscript might be improved.
In general, figures should not precede their first mention in the text. Otherwise it is not obvious why the figure is there.
Thanks for kind suggestion. The positions of figures in manuscript were relocated.
In the paragraph starting on line 105, authors describe the fabrication of the MIS capacitors. They write, “Finally, a silver (Ag) electrode of 100 nm was deposited…”. According to Figure 1a, the MIS capacitor as an Al top electrode, not Ag. Ag is used for the NVM. This should be clarified. Also, why use Al for the MIS device but Ag for the NVM? Explain.In addition, the ion mass spectroscopy for calculating the ratio of Ga was lacked..
Thanks for kind suggestion. The position of the figure is relocated.
The mark and caption for Fig. 1(a) were mispresented. The electrode of C-V device is silver (Ag), the same as the electrode of NVM device.
In our experiment, Ga was not included in the active layer, but we used Ga-free ITZO layers using a sputter target of In2O3:SnO2:ZnO (=1:1:1); we did not need a mass spectrometer because we used a target that had already been determined.
In Figure 1, what are the reasons for the layer thicknesses used? What was the doping concentration of the Si wafer, for MIS as well as the gate for the NVM?
Thanks for kind suggestion. We added the statements as;
“… The layer thicknesses of the device are very important for the proper operation. Especially, the thickness of the blocking layer can directly affect the internal electric field and the amount of the trapped charges. We used the thickness of each layer optimized through the previous experiments [14]. …”
and we modified the statement as;
“… on a n-type single crystalline Si substrate of about 1-5 ΩNaN resistivity …”
Which specific semiconductor parameter analyzer was used in these measurements?
We added the statements as;
“The current-voltage and capacitive-voltage characteristics for the fabricated devices were measured by a semiconductor parameter analyzer (EL420C) and an impedance analyzer (Hewlett Packard 4192A LF), respectively.”
In the paragraph starting on Line 166, authors state that –V results in accumulation and a +V results in depletion on these CV curves. It seems to me that the opposite must be true. First of all, if these are n-type substrates, then positive voltages normally yield accumulation. This is also apparent from the CV curves. +VG is in accumulation, – VG is in depletion and inversion. In the next sentence the authors seem to change their statement of what is accumulation and what is depletion. This should be clarified.
As your comments, there is an error in the sign of the applied voltage. We correct it in the revision.
On line 174 authors refer to “seriously-connected capacitors…”. Do authors mean “series-connected”?
It was a typo; we correct it as “series-connected” as your comment.
In Figure 3, what do the different colors mean? This is not made clear initially. Authors state, “As the sweep voltage range increases from – 5 ~ 5 V to – 15 ~ 15 V, “. This changing sweep range is not discussed earlier, and just bringing it up abruptly is unclear. I think I know what Authors are trying to do in this experiment, but the reader should not have to work that hard to figure this out. Explain it to the reader before presenting the results.
We added descriptions in the legend of Fig. 3.
On line 200, a paragraph ends where the authors make the very important point of the differences between CV results for GR0.5, GR2, and GR6. But, they leave the reader hanging. Finish the thought, tie up the paragraph, and tell the reader what this means. We can figure it out, but the authors should tell the reader the significance if there is any.
Thanks for kind suggestion. We added to tie up the paragraph.
“… As shown in Figure 2, the charge trapping layer with more Si phases can increase the ΔVFB (memory window) to more clealy distinguish memory states. The SiOX layer grown at the highest SiH4:N2O gas ratio (GR6) condition can capture the greatest amount of charge with a large number of Si phases.”
Since erasing requires light assistance, this seems to make this device less practical, which the authors mention at the end of the paper. What prospects are there for developing a method for effectively erasing these NVM devices without light? This is a very important questions to answer.
Thanks for kind suggestion. Our suggestions are added at the end of p.8 as;
“A bandgap engineering for the layer structure, optimizing the conduction/valance band offset between the tunneling layer and the charge trapping layer, can reduce tunnel barrier and improve the erasing operation, or the excitations of the trapped charges which can be done by local thermal heating or electro-magnetic resonating terminals can promote the tunneling process without the external light source.”
Reviewer 2 Report
The manuscript "The characteristics of transparent non-volatile memory devices employing Si-rich SiOX as a charge trapping layer and indium-tin-zinc-oxide" reports on the fabrication of transparent non-volatile memory of a bottom gate thin film transistor for the integrated logic devices of display applications.
From a general point of view, the topic of the manuscript is interesting and worth of investigation. Even if the topic is studied by several years, the authors are able in reporting some new and interesting data allowing original insights. The manuscript is clear, well-written and well-organized. Figures are clear and appealing.
The introduction clearly states the aim of the work and sharply inserts the work within a general framework. The experimental section is clear, detailed and complete. The experimental analysis are strongly founded. The experimental results are interesting and really promising. The experimental results are discussed on the basis of strongly quantitative analysis and considerations based on general reliable physical concepts.
On the basis of these consideration, I think that this a nice manuscript which allows new and interesting insights. So, surely, it deserves publication.
However, before publication, I suggest to the authors some clarifications:
1) What about the tunneling mechanism? Could it be associated to some specific process? Direct Tunneling? Defect-assisted tunneling? Fowler-Nordheim? Single-tunnelling effects?
For example, see and comments on the basis of: Advances in Materials Science and Engineering 2014, 578168 (2014); Appl. Phy. Lett. 89, 263108 (2006); etc.
2) A critical comparison to other technologies approaches is strongly suggested. See: Materials 7, 5117 (2014).
3) What about the uniformity (roughness) of the tunneling layer?
Author Response
The manuscript "The characteristics of transparent non-volatile memory devices employing Si-rich SiOX as a charge trapping layer and indium-tin-zinc-oxide" reports on the fabrication of transparent non-volatile memory of a bottom gate thin film transistor for the integrated logic devices of display applications.
From a general point of view, the topic of the manuscript is interesting and worth of investigation. Even if the topic is studied by several years, the authors are able in reporting some new and interesting data allowing original insights. The manuscript is clear, well-written and well-organized. Figures are clear and appealing.
The introduction clearly states the aim of the work and sharply inserts the work within a general framework. The experimental section is clear, detailed and complete. The experimental analysis are strongly founded. The experimental results are interesting and really promising. The experimental results are discussed on the basis of strongly quantitative analysis and considerations based on general reliable physical concepts.
On the basis of these consideration, I think that this a nice manuscript which allows new and interesting insights. So, surely, it deserves publication.
However, before publication, I suggest to the authors some clarifications:
1) What about the tunneling mechanism? Could it be associated to some specific process? Direct Tunneling? Defect-assisted tunneling? Fowler-Nordheim? Single-tunnelling effects?
For example, see and comments on the basis of: Advances in Materials Science and Engineering 2014, 578168 (2014); Appl. Phy. Lett. 89, 263108 (2006); etc.
Thanks for kind suggestion. We added mention in the paper.
“Various tunneling mechanisms have been suggested, such as direct tunneling, Fowler-Nordheim (FN) tunneling and channel hot electron (CHE) tunneling processes [19]. In our work, the ultra-thin tunneling layer less than 4 nm is used, and the energy barrier (ΦB) between ITZO and SiOXNY is ~2 eV, so we expect that a direct tunneling process can be dominantly involved. We note that specifying one of these processes is difficult because these processes can occur together. So it can be done in future studies.”
2) A critical comparison to other technologies approaches is strongly suggested. See: Materials 7, 5117 (2014).
We appreciate the reviewer’s suggestion of a ref. paper, but the paper was mainly about Si channel. In our work, we used the semiconductor oxide channel. We are afraid that referencing that paper could not be proper for our case.
3) What about the uniformity (roughness) of the tunneling layer?
Thanks for kind suggestion. Because every thin films used in our NVM was amorphous phase, the uniformity is not important, and we can do the experiments reproducible without great difficulty.
Reviewer 3 Report
Please see the comments below:
This manuscript reports a novel non-volatile memory fabricated using a gallium free In-Sn-Zn-oxide as the active channel layer and SiO2/Si- rich SiOx/SiOxNy as the gate insulator. The study is novel and significant. The following areas should be addressed:
“The increase of states in the charge trapping layer was investigated by a fourier transform infrared (FT-IR) spectroscopy, and the expansion of the memory window was confirmed by capacitive-voltage (C-V) measurement.” – What type of FTIR was used? What was the model number? Was the measurement done using an ATR stage? Details should be provided for the sample preparation and measurement.
“To characterize and optimize Si-rich SiOx layer, metal – insulator – silicon (MIS) capacitors were fabricated as shown in Figure 1(a); the stack of aluminum / silicon dioxide (SiO2) / Si-rich SiOX / silicon oxynitride (SiOXNY) layer was deposited by inductive coupled plasma chemical vapor deposition (ICP-CVD) on a n-type single crystalline Si substrate.” – Please include the model and company of the ICP-CVD used. A photo of the fabricated device should be included along with the schematic.
“The SiOX layers showed the high absorption of Si-O bending peaks (or defects known as non-bridging oxygen hole center, NBOHC) at 860 cm-1 [14]” – Was there any increase in the Si-O bond angle that could be detected from the peak shift?
In particular, SiOx having a high Si content shows a remarkably large number of Si-H bonds as well as HSi3O, HSi2O2, H2SiO2, and HSiO3 at around 2000 ~ 2300 cm-1 [15-16], and as the GR increases, the absorptions around 2000-2300 cm-1 also increases.” – The increase in absorption around 2000- 2300 cm-1 should be quantified.
“To investigate light induced erasing behavior, the optical bandgap was measured by UV-visible spectra. The optical bandgap of ITZO and SiOXNY, SiOX, is 3.05 ~ 3.15, 5.1 and 2.3, respectively.” – The unit for the bandgap must be included. The Uv-vis spectra and the extrapolation to determine the optical band gap should be shown as a separate figure.
“Using the optimized Si-rich SiOx charge trapping layer, a bottom gate charge trapping NVM with ITZO layer was fabricated as shown in Figure 1(b).” – What composition of ITZO layer was used for the fabrication?
Author Response
This manuscript reports a novel non-volatile memory fabricated using a gallium free In-Sn-Zn-oxide as the active channel layer and SiO2/Si- rich SiOx/SiOxNy as the gate insulator. The study is novel and significant. The following areas should be addressed:
“The increase of states in the charge trapping layer was investigated by a fourier transform infrared (FT-IR) spectroscopy, and the expansion of the memory window was confirmed by capacitive-voltage (C-V) measurement.” – What type of FTIR was used? What was the model number? Was the measurement done using an ATR stage? Details should be provided for the sample preparation and measurement.
The transmittance mode (Bruker, IFS-66/S) FT-IR was used for the charge trapping layer evaluation.
“To characterize and optimize Si-rich SiOx layer, metal – insulator – silicon (MIS) capacitors were fabricated as shown in Figure 1(a); the stack of aluminum / silicon dioxide (SiO2) / Si-rich SiOX / silicon oxynitride (SiOXNY) layer was deposited by inductive coupled plasma chemical vapor deposition (ICP-CVD) on a n-type single crystalline Si substrate.” – Please include the model and company of the ICP-CVD used. A photo of the fabricated device should be included along with the schematic.
The ICP-CVD used in our experiment was custom ordered by a Korea domestic company; the model and company name were insignificant for the readers of the journal.
We added an optical microscopy image of the device in Figure 5(a).
“The SiOX layers showed the high absorption of Si-O bending peaks (or defects known as non-bridging oxygen hole center, NBOHC) at 860 cm-1 [14]” – Was there any increase in the Si-O bond angle that could be detected from the peak shift?
Thanks for your comments. We confirmed that the Si-O bending peaks as well as Si-H peaks were not shifted.
In particular, SiOx having a high Si content shows a remarkably large number of Si-H bonds as well as HSi3O, HSi2O2, H2SiO2, and HSiO3 at around 2000 ~ 2300 cm-1 [15-16], and as the GR increases, the absorptions around 2000-2300 cm-1 also increases.” – The increase in absorption around 2000- 2300 cm-1 should be quantified.
As your comments, the measured FT-IR result around 2000 ~ 2300 cm-1 are added in Figure 2 as;
“To investigate light induced erasing behavior, the optical bandgap was measured by UV-visible spectra. The optical bandgap of ITZO and SiOXNY, SiOX, is 3.05 ~ 3.15, 5.1 and 2.3, respectively.” – The unit for the bandgap must be included. The Uv-vis spectra and the extrapolation to determine the optical band gap should be shown as a separate figure.
Thanks for your comments. We added the units in the text, and the UV-vis spectra were added in Figure 6(a) as;
“Using the optimized Si-rich SiOx charge trapping layer, a bottom gate charge trapping NVM with ITZO layer was fabricated as shown in Figure 1(b).” – What composition of ITZO layer was used for the fabrication?
The ITZO target for the fabrication was composed of In2O3:SnO2 : ZnO (= 1:1:1), so, the composition of In:Sn:Zn ratio is 1:1:1.

Round 2
Reviewer 3 Report
Accept in the present form.